# Assessing Vegetation Ecosystem Resistance to Drought in the Middle Reaches of the Yellow River Basin, China

**DOI:** 10.3390/ijerph19074180

**Published:** 2022-03-31

**Authors:** Xiaoliang Shi, Fei Chen, Hao Ding, Mengqi Shi, Yi Li

**Affiliations:** College of Geomatics, Xi’an University of Science and Technology, Xi’an 710054, China; xiaoliangshi@xust.edu.cn (X.S.); 19210061018@stu.xust.edu.cn (H.D.); 20210061026@stu.xust.edu.cn (M.S.); 20210061035@stu.xust.edu.cn (Y.L.)

**Keywords:** drought characterization, correlation coefficient, vegetation resistance, SPEI, NDVI

## Abstract

The frequency and intensity of droughts are increasing in many parts of the world as a result of global climate change and human activity, posing a serious threat to regional ecological security. The climate in the middle reaches of the Yellow River Basin (MRYRB) has been warm and dry in recent years, with frequent droughts. In order to investigate the temporal trend of drought, and reveal the resistance of vegetation to drought in the MRYRB, this study used remotely-sensed vegetation index products (MODIS-NDVI and SPOT-NDVI) and the standardized precipitation evapotranspiration index (SPEI). The results indicated that: (1) drought intensity showed a weak upward trend in the study area from 2000 to 2018, with linear growth rates of SPEI at temporal scales of 1, 3, 6, 9 and 12 months of −0.002, 0.0034, 0.0198, 0.0234, and 0.0249, respectively; (2) drought was positively correlated with vegetation in most areas (97.6%), and vegetation was most affected by drought on long-term time scales (9 and 12 months); (3) with the extension of drought, vegetation resistance index decreased, then gradually recovered after the end of the drought. Forest had the longest resistance duration of 260 days, while grassland and cultivated land had resistance durations of only 170 days. This study adds to the understanding of vegetation’s ability to withstand drought, and these findings provide evidence to support drought response in the MRYRB.

## 1. Introduction

Drought is a condition relative to some long-term average condition of balance between rainfall and evapotranspiration in a particular area [1]. Human activities cause an increase in the concentration of carbon dioxide and other heat-trapping gases in the atmosphere, which in turn leads to an increase in surface temperature and global warming, and the frequency and intensity of droughts will continue to increase by the end of the 21st century, especially in arid regions [2,3]. The composite drought hot spot map by Samiul et al. [4] draws attention to the negative effects of drought, which not only restrict social and economic development but also cause a series of ecological and environmental problems such as water shortages, soil degradation, sandstorms, and desertification [5]. Therefore, it is critical to carry out drought assessments and investigate drought evolution and response relationships as a result of climate change.

Vegetation provides important ecosystem services related to human well-being, biodiversity and carbon cycling, and is an important component of terrestrial ecosystems [6,7]. Vegetation is characterized by adaptation to its environment, and therefore, close monitoring of vegetation spectral characteristics is of ecological importance [8,9]. Studies based on field experiments and remote sensing methods have shown that drought has a significant negative impact on vegetation growth, and is affected by drought duration and intensity [10,11]. During a drought, vegetation growth slows down or the vegetation dies, which threatens regional ecological security [12,13]. Therefore, understanding the response of vegetation to drought is critical in the context of climate change, and further understanding of how drought affects vegetation ecosystems can help people cope with rising drought stress and provide scientific guidance for water resource allocation and drought prevention [14].

Quantifying drought characteristics helps to assess drought events, and drought indexes are one of the most effective tools for identifying and describing drought events in a timely manner [15]. The standardized precipitation evapotranspiration index (SPEI) can be used to define the severity of drought at various time periods, to efficiently monitor drought and to quantify drought episodes [16]. SPEI considers both temperature and precipitation and is able to detect the effects of evapotranspiration and temperature changes on global warming [17], which is widely used by climatologists in climatology research [18,19]. For example, SPEI was analyzed by Nejadrekabi et al. [20] using the Getis-Ord-Gi statistics to determine the SPEI values that form clusters and identify areas at high risk of drought; Sharma et al. [17] used geospatial techniques to generate the SPEI drought map of Tripura, Northeast India in 1985 to describe the severity of the drought. The normalized difference vegetation index (NDVI) is a feedback variable in the ecosystem that is used to indicate vegetation growth and vegetation cover changes. It is one of the most well-known vegetation indexes and is produced from observed reflectance in the red and near-infrared sections of remotely-sensed spectra [21,22,23]. Currently, the most easily accessible global remote sensing NDVI data are the Moderate Resolution Imaging Spectroradiometer NDVI (MODIS-NDVI) and the SPOT-VEGETATION NDVI (SPOT-NDVI), both of which have been widely used. The MODIS-NDVI has the advantages of high spectral resolution and high spatial resolution, while the SPOT-NDVI is specially designed for vegetation and surface observation and has a higher temporal resolution.

In arid and semi-arid areas, response relationships between vegetation and drought based on correlation analysis have been assessed using NDVI and SPEI. For example, Zhao et al. [19] investigated the response of vegetation at different time scales to seasonal water stress on the Loess Plateau, China, using maximum Pearson correlations. Xu et al. [24] examined drought responses and drought resistance of vegetation in northern China among different biome types and climatic zones using Spearman correlation analysis. These two studies indicated that drought inhibits vegetation growth. However, the relationship between vegetation and drought cannot be clearly demonstrated with a simple correlation analysis, and the response mechanism of vegetation to drought has to be investigated further.

Vegetation can improve its resistance to drought by changing its structure and physiological characteristics in a process of long-term adaptation to the environment. Vegetation resistance index is used to assess vegetation’s ability to maintain its original characteristics during droughts, which can be measured using a vegetation index. Vegetation with higher resistance can effectively mitigate the harmful effects of drought on an ecosystem [25,26]. However, most studies to date have been limited to the calculation of simple correlations between drought indexes and vegetation indexes in order to assess ecosystem resistance to drought, and these have focused primarily on spatial distributions of correlations [27,28]. Vegetation resistance is mostly determined by the nature of climate anomalies, but the various degrees of drought disturbance and the delayed effects of drought on vegetation have not yet been fully taken into account in current studies. Vegetation resistance is influenced by the physiological structure of vegetation types as well as by drought duration and intensity, but there is little research on long-term changes in resistance across different vegetation types. Therefore, this study uses the high temporal resolution of the SPOT-NDVI to study the change in vegetation resistance on a 10-day scale in relation to a specific drought event.

Due to the rapid growth of the population and economy, the ecological environment in the middle reaches of the Yellow River Basin (MRYRB) is fragile, with a long-term decrease in river runoff and a rise in average temperature, the environment has gradually changed from a natural environment to a human-influenced environment which is very prone to drought [29]. In recent years, the frequency and intensity of drought events in the MRYRB have shown increasing trends and droughts have become the main factor restricting the further development of the MRYRB, manifested primarily in regional ecological security, social and economic development, and industrial and agricultural production [30,31,32].

Thus, the aims of this study are: (1) to reveal the temporal characteristics of drought, and vegetation responses to drought, in the MRYRB on different time-scales; (2) to assess the temporal change of vegetation resistance index during the development of a drought event and determine the resistance duration to drought based on vegetation characteristics; (3) to discuss the causes for the spatial heterogeneity of the response relationship between vegetation and drought. It is hoped that the results can strengthen disaster management in terrestrial ecosystems and provide a theoretical basis for responding to climate change in the MRYRB.

## 2. Materials and Methods

### 2.1. Study Area

The MRYRB spans between 32°–42° N and 104°–112° E, with a total area of approximately 3.44 × 105 km^2^ (Figure 1). It belongs to the arid, semi-arid, and semi-humid climatic zones, with four distinct seasons. Precipitation and temperature have significant seasonality and show an uneven distribution in both time and space. The average annual precipitation is about 480 mm, with more than 70% falling from June to September and the average annual temperature ranges from 8 °C to 14 °C. Most areas of the MRYRB contain loess soil, with weak erosion resistance and high infiltration capacity. The main types of coverage are forest, grassland and cultivated land, accounting for 20.4%, 35.1%, and 37.1% of the total area, respectively (Figure 1a). Since the implementation of the Three-North Shelterbelt Project in the 1990s, the regional ecological environment has improved to a certain extent. Restricted by the inherent natural climatic conditions, the frequency of drought disasters in the study area is increasing. In addition, relevant studies have shown that under the background of RCP 8.5, longer-lasting droughts will occur in the future in the MRYRB [33,34].

### 2.2. Data

#### 2.2.1. Remote Sensing Data

MODIS-NDVI and SPOT-NDVI datasets were used in this study. The time series of the monthly MODIS-NDVI dataset (MOD13A3) had a 1 km resolution and was downloaded from the website of the US National Aeronautical Space Agency (NASA) (https://ladsweb.modaps.eosdis.nasa.gov, accessed on 1 May 2021). The data covered the period from 2000 to 2018. The 10-day maximum SPOT-NDVI (SPOT-VEGETATION NDVI) synthesis images at 1 km spatial resolution were obtained from the Copernicus Global Land Service (https://land.copernicus.eu/global/products/ndvi, accessed on 29 January 2022) for 2010–2012, which has removed the effects of clouds, ice and snow [35]. The monthly MODIS-NDVI data was utilized to demonstrate the distribution of water stress and vegetation correlations, whereas the 10-day SPOT-NDVI data was employed to characterize the vegetation resistance index temporal variation.

A 1 km land-cover classification product from 2018 was obtained from the Resource and Environment Science and Data Cloud Platform (http://www.resdc.cn/, accessed on 1 March 2021). The dataset was built using high-resolution remote sensing satellite imagery data, unmanned aerial vehicles and ground survey observation systems. The comprehensive evaluation accuracy of the first level categories of this dataset was above 93% [36]. The obtained land-cover data are summarized and reclassified into 6 categories: grassland, forest, cultivated land, construction land, water body and others (Figure 1a). The study in this paper focuses on grassland, forest, and cultivated land.

A geomorphic type of data was from the Chinese 1:1,000,000 Geomorphic Atlas, which is divided into four primary types: plain, tableland, hill and mountain (Figure 1b).

A digital elevation model (DEM) with 90 m resolution was provided by the Geospatial Data Cloud site, Computer Network Information Center, Chinese Academy of Sciences (http://www.gscloud.cn, accessed on 25 October 2021) (Table 1).

#### 2.2.2. Standardized Precipitation Evapotranspiration Index (SPEI)

SPEI product datasets for the time scales of 1, 3, 6, 9 and 12 months came from the Climate Research Unit (CRU) (https://digital.csic.es/handle/10261/202305, accessed on 25 October 2021) with a spatial resolution of 0.5° × 0.5° for the period 2000–2018, representing short (1 and 3 months), medium (6 months), and long (9 and 12 months) time scales, respectively. This multi-scalar metric represents either a water surplus or deficit for a given month [37]. Studies have confirmed the reliability of this dataset and have previously applied it to drought-related research [38]. The data set was resampled to match the 1 km spatial resolution of the vegetation indices using the Resample Tool of the ArcGIS software package.

### 2.3. Methods

#### 2.3.1. Vegetation Resistance

As discussed above, the vegetation resistance index (Ω) is one of the most widely used approaches to describe the response of vegetation to climate disturbances and is suitable for ecosystems with a high sensitivity to stress [39,40]. It is calculated as follows:(1)$$\Omega =\frac{\bar{Y_{n}}}{\left|Y_{e}-\bar{Y_{n}}\right|}$$
where $\bar{Y_{n}}$ and *Y_e_* represent the mean value of an NDVI during normal years (mean across all non-drought years) and the NDVI value during the occurrence of drought, respectively. The maximum and minimum of Ω are ∞ and 0. Ecosystems with greater Ω are more resistant than others. The resistance index used in this study is unitless, and thus can be directly comparable across biomes with varying productivity levels.

The time from the start of the drought to the time of the disturbance of vegetation was recorded as the vegetation resistance duration to drought. Its calculation formula is as follows:(2)$$T=T_{\beta }-T_{\alpha }$$
where *T* represents the vegetation resistance duration, *T_α_* and *T_β_* represent the time when the drought started and vegetation was disturbed, respectively. The non-drought value was defined as the baseline to assess drought impact against, i.e., when an NDVI continuity became lower than the baseline, this indicated that a drought had disturbed vegetation.

#### 2.3.2. Analysis Methods

To calculate the trend in variable using the linear tendency method [41]. In this study, the variable is SPEI. The formula is as follows:(3)y=a+bx
where *y* is the time series of SPEI, *x* is the corresponding time series; the regression coefficient *b* represents the linear trend, and *a* is a constant. The value of *b* greater than 0 implies an increasing trend.

Pearson correlation (PC) is an index to measure the degree of correlation between two variables [42]. In this study, the annual correlation coefficient of SPEI and NDVI in image pixels at different time scales was calculated, and the spatial distribution characteristics of the correlation coefficient between vegetation and drought were analyzed. Its calculation formula is as follows:(4)$$R=\frac{\sum_{i=1}^{n}\left[\left(x_{i}-\bar{x}\right)\left(y_{i}-\bar{y}\right)\right]}{\sqrt{\sum_{i=1}^{n}{\left(x_{i}-\bar{x}\right)}^{2}}\sqrt{\sum_{i=1}^{n}{\left(y_{i}-\bar{y}\right)}^{2}}}$$
where *R* represents the PC, *x_i_* and *y_i_* represent the NDVI and SPEI values in the *i*th year, $\bar{x}$ and $\bar{y}$ are the mean values of SPEI and NDVI. The maximum correlation coefficient was used in order to avoid a small number of outliers as a result of interference.

In arid and semi-arid areas, in order to highlight the vegetation situation and eliminate the interference of abnormal factors, the Maximum Value Composite (MVC) method was adopted to obtain the annual NDVI image [43], and its calculation formula is as follows:(5)$$M=Max\left(NDVI_{i}\right)$$
where *M* is the maximum NDVI, *NDVI_i_* is the NDVI in the *i*th month. The maximum NDVI image can reflect the annual variation of vegetation more clearly.

In order to detect overall trends in the NDVI from 2000 to 2018 in our study area, a least-squares linear regression model was applied and fitted to the multi-year NDVI dataset [14], i.e.,:(6)$$Slope=\frac{n\times \sum_{i=1}^{n}i\times NDVI_{i}-\left(\sum_{i=1}^{n}i\right)\left(\sum_{i=1}^{n}NDVI_{i}\right)}{n\times \sum_{i=1}^{n}i^{2}-{\left(\sum_{i=1}^{n}i\right)}^{2}}$$
where *n* is the number of years (*n* = 19), and *i* is an integer ranging from 1 to *n*. The *NDVI_i_* represents the NDVI for the *i*th year, while the *Slope* represents the trend of the NDVI during 2000–2018. If the value of *Slope* is greater than zero, this implies that vegetation has improved.

## 3. Results

### 3.1. Drought Characterization in the MRYRB

The results of the linear trend analysis showed that drought severity in the MRYRB has increased slightly from 2000 to 2018 (Figure 2). The linear growth rates of SPEI at time scales of 1, 3, 6, 9 and 12 months were −0.002, 0.0034, 0.0198, 0.0234, and 0.0249, respectively, all of which passed the significance test (*p* < 0.01). With the exception of a time scale of 1 month (SPEI-1), SPEI on other scales exhibited an increasing tendency, and the trend increased with increasing time scale.

In terms of individual years (Figure 2), in 2003 there was a significant rise in precipitation and the entire region was free of drought. However, all other years experienced varying degrees of drought. The drought changes in the 2000–2018 basins alternate, and the year pairs with the most noticeable dry-wet alternation were 2004–2005, 2009–2010, and 2011–2012. If the observed trends continue, drought duration and severity will continue to increase in the MRYRB. This is consistent with the observation of Liu et al. that warm-dry is the dominant climate feature in the region, and that alternating high-frequency warm-humid, warm-dry, and warm-normal changes will become the mainstream climate feature in the 21st century under the projected conditions of climate change scenarios RCP2.6, RCP4.5, and RCP8.5 [33].

### 3.2. Spatial Distribution of the Correlation between NDVI and SPEI

Most image pixels in the MRYRB had a positive correlation between MODIS-NDVI and SPEI, accounting for 97.6% of the study area. Correlations were particularly high in Gansu Province and the Ningxia Hui Autonomous Region, where correlation coefficients were greater than 0.5 (Figure 3a). Only 2.4% of the pixels in the study area exhibited negative correlations, and these were mainly located in the Guanzhong Plain and in some regions of Henan Province. These regions have a higher level of economic development and urbanization, they are heavily influenced by human activity, and climate change has less of an impact on such urbanized land. In general, the vegetation showed a positive response to drought in the west and center of the MRYRB, but a weak negative response to drought in other regions, and our results match those of Zhang et al. [44].

The response sensitive areas of NDVI to SPEI at temporal scales of 1, 3, 6, 9 and 12 months accounted for 2.4%, 3.5%, 23.8%, 31.0%, and 39.3% of the study area, respectively, indicating that the annual NDVI of MRYRB was most affected by SPEI at 12 months (SPEI-12) (Figure 3b). The areas sensitive to 3 and 6 months were mainly distributed in grassland and cultivated land areas with significant growth cycles. The time scale at which the forest was sensitive to drought stress was mainly 9 months and above, and the forest was not susceptible to short-term water stress. The correlation coefficients between NDVI and SPEI in the eastern and central parts of the study area were lower than those in the western part, but the corresponding time scales in the eastern and central regions were higher than those in the western part of the study area.

### 3.3. Vegetation Resistance during Typical Drought Events

The winter-spring drought from October 2010 to February 2011 was selected as a typical drought event by the Bulletin of Flood and Drought Disasters in China [45]. This drought involved 8 provinces and nearly 100,000 ponds. The precipitation decreased by more than 50% from October onwards, and terrestrial water storage declined. Drought quickly spread in mid-November, reaching a climax in early February the following year, and affecting an area of around 7434 hectares. As temperatures rose and precipitation increased around mid-February, the drought began to ease. The drought impact is quantified by comparing the difference in vegetation activity between drought years and baseline conditions [46]. Considering the climate change in the MRYRB in the past two decades, the period 2011–2012 was selected to represent conditions without drought and was used as the baseline period representing normal conditions.

#### 3.3.1. Variation Trend of the Vegetation Resistance Index

The lowest limit of the vegetation resistance index was determined by finding the minimum resistance of different vegetation types (Figure 4). The temporal variations of the resistance index in three vegetation ecosystems (forest, cultivated land, grassland) were not consistent. Forest resistance began to fall in the second month of the drought, remained at about 0.5 from early December to early February, and then began to rise in mid-February until it returned to normal while grassland and cultivated land recovered faster. Similar to forest resistance, grassland resistance fluctuated in the second month of the drought, remained at 0.5 from early December to mid-March, then began to rise in late March. The recovery duration was longer than that of the forest, and the recovery rate was slower. At the start of the drought in October, the resistance of cultivated land fluctuated as it was influenced by human activity, and the resistance did not always reduce. Resistance unexpectedly increased in late November, fluctuated, reduced and then gradually returned to normal by mid-March.

The resistance index of cultivated land was lower than that of forest and grassland during the drought (Figure 4). Forest had the lowest resistance index of 0.279 in the middle of December, grassland had the lowest resistance index of 0.169 in the middle of January, and cultivated land had the lowest resistance index of −1 in late January and early February. The maximum and minimum values for cultivated land occurred later than those of forest and grassland. The sudden rise in resistance for cultivated land in late November may have been caused by the artificial irrigation of crops. As February is the green season of winter wheat in the MRYRB, the continuous growth of water demand led to vegetation resistance on cultivated land reaching its lowest value. Due to the dry winter environment, limited vegetation coverage and lower NDVI values on cultivated land due to winter and spring droughts, drought resistance was unfavorable during this period. The resistance of forest and grassland fluctuated dramatically at the end of the drought period, whereas the resistance of cultivated land stayed low and then recovered slowly in the non-growing season with increasing soil evaporation. During a drought event, the vegetation resistance index generally fluctuated, even diminished, and then gradually returned to normal after the peak of the drought.

#### 3.3.2. The Vegetation Resistance Duration to Drought

Extraction of drought areas based on SPEI images, and comparison of NDVI in drought areas to baseline show that drought had little effect on the overall NDVI seasonal variation. However, the NDVI showed a dramatic decrease during the middle and late stages of drought development compared with normal years (Figure 5). The lowest NDVI value occurred in the later winter of 2010. The NDVI of grassland and cultivated land exhibited a similar decreasing pattern during drought but the magnitude of decrease was larger than forest. In late November and early January, the NDVI of grassland was lower than the baseline. By late March, the NDVI of grassland had dropped significantly compared to the same time the previous year. Drought caused severe damage to grassland, and the grassland was able to withstand the drought event for 170 days (1 October 2010–11 March 2011).

The NDVI of cultivated land during the drought period was lower than corresponding baseline values for the first time in late October, lower than the baseline in mid-November and early January again, and significantly lower than the baseline from late March onwards. Hence the resistance duration was 170 days (1 October 2010–11 March 2011), which was consistent with the resistance duration of grassland, but the time of fluctuation was earlier than that of grassland, and the frequency of fluctuation was higher than that of grassland. As the root systems of grassland and cultivated land are shallow and water storage capacity is low, the resistance duration to drought was reduced.

In mid-November, early January, mid-February, and late February, the NDVI of forest varied and was lower than the baseline. After experiencing a chronic water shortage environment, NDVI values began to be much lower than baseline values from mid-June, and resistance duration was 260 days (1 October 2010–11 June 2011), which was longer than for grassland and cultivated land. With established roots, a forest may be more able to maintain development by extracting soil water when precipitation decreases, which result in smaller variations of NDVI during the drought. Therefore, forest resistance duration was longer.

## 4. Discussion

### 4.1. Response of Different Geomorphic Types to Drought

According to previous studies, the response of vegetation to drought is influenced by geomorphic types, and hence uneven spatial distribution is evident [19,47,48]. The MRYRB runs through the heart of the Loess Plateau, and there are numerous tributaries due to tectonic movement, the geomorphic types of the region feature considerable spatial variability. Plain, tableland, hills and mountains make up the majority of the basin. The response of vegetation to drought is influenced by changes in lithology, soil, and hydrology among different geomorphic types [49,50]. Correlation coefficients between vegetation and drought were calculated for different geomorphic types (Figure 6a). The vegetation in hill regions was the most drought-susceptible. Except for a few aberrant values, the majority of the correlation coefficients between vegetation and drought in this area were between 0.28 and 0.44, and all correlation coefficients were positive. The plain region had the lowest correlation coefficients, with a maximum of 0.73 and a minimum of −0.19. The maximum correlation coefficients between vegetation and drought were similar for both mountain and plain regions. Each geomorphic type was more vulnerable to medium and long-term water deficits when considering the response time scale (Figure 6b). However, plain and tableland also show reactions to short-term water deficits.

The multi-pore soil structure, vertical joint development, robust permeability and rich calcium carbonate properties of the loess soil in the MRYRB make it more suited for vegetation growth and a wide number of weathering products produce thicker soil and have a high capacity for water storage. The small size and large surface area of Loess soil particles improve their ability to absorb ions as well as improve their water storage capacity, lowering vegetation’s reliance on precipitation. As a result, the MRYRB was mainly affected by the SPEI on a 12-month time scale.

Owing to the high altitude in mountain and hill regions, the response of vegetation growth to drought is mainly affected by surface temperature, snow cover, and energy [51,52]. Temperature plays a decisive role in vegetation growth, and energy is a key limiting factor. A large amount of snow can result in possible excess water and reduced soil water loss [53]. As a result, these regions may be vulnerable to the twin effects of water and energy scarcity, and they can withstand short-term water shortages.

The response relationship between vegetation and drought in low-altitude environments such as plains and tablelands is influenced by population density. Human activities such as urbanization and grazing degrade vegetation biomass, vegetation cover and several soil functional indicators, increasing drought sensitivity [54]. As a result, a short-term drought will also jeopardize the region’s ecological system’s security.

### 4.2. Divergent Resistance of Different Vegetation Types

The maximum correlation coefficients between NDVI of various vegetation types and SPEI at different time scales were calculated, and the difference in response for the various vegetation types was investigated (Figure 7). Drought had a similar impact on both cultivated and grassland areas. In terms of response time, grassland and cultivated land had the strongest responses to medium time scales (6 months), with the highest correlation coefficients of 0.793 and 0.825, respectively. The maximum correlation coefficient between forest NDVI and SPEI at a time scale of one month was only 0.591, whereas the maximum correlation coefficient between forest NDVI and SPEI at a 9-month time-scale was 0.839. In short-term (1 and 3 months) drought events, the correlation coefficient between NDVI and SPEI of grassland and cultivated land was higher than that of the forest, and a short-term water deficit could easily affect the growth of cultivated land and grassland. Forest was more sensitive to long-term droughts, and the longer the time scale, the more sensitive the forest was to drought.

It has been shown that a forest can experience higher drought risks than grassland and cultivated land. The endurance capacity of vegetation to drought events is linked to both the degree of the drought damage and the physiological properties of the vegetation. In a water-scarce environment, vegetation responds to drought primarily by collecting water from the soil through its roots. We found that grassland NDVI showed a gradual decrease compared to baseline during the drought event (Figure 5b). Grassland is shallow-rooted vegetation with periodic growth, which is consistent with observed cycle NDVI variation. A leaf with a short life has a faster photosynthetic rate than a leaf with a longer life, and such a leaf mostly obtains water from surface or shallow soil water. Precipitation is a crucial element impacting grassland response to drought [55]. A grassland ecosystem may respond quickly to proper growth conditions, making it vulnerable to short-term meteorological droughts [56].

In this study, there were obvious spatial differences within grassland ecosystems (Figure 3). The southern part of the MRYRB contains sparse grasslands and woody sparse grasslands, which experience hot, seasonal arid climate conditions. Such vegetation will experience growth for long periods with a lack of water every year. Hence, they have developed cold and drought tolerant characteristics and were slightly less responsive to drought than the general grassland [57].

The cultivated land showed a strong drought response to SPEI at a 6-month time-scale and its resistance was the worst (Figure 4). In the winter-spring drought of 2010–2011, the cultivated land resistance duration was 170 days, this indicates that it was more sensitive to climate change (Figure 5c). The cropping ecosystem in the MRYRB consists of two crops a year, and the crops are mainly wheat, corn, and soybeans, whose roots cannot access deep groundwater and whose water use efficiency is affected by human activities. Drought resistance of cultivated land is affected by a number of field management practices such as irrigation and fertilization strategy and new planting technology, which influence the water absorption capacity of crop roots and leaf photosynthesis and transpiration rates [58]. Crop growth is closely related to soil texture and light and heat conditions. In areas with a high level of urbanization, water is no longer the most important factor limiting vegetation growth. However, precipitation is still the main limiting factor of crop growth in dry areas. The growth status of grassland and cultivated land is closely related to the intensity and duration of drought. In the monitoring of drought events, attention should be paid to the status of low and shallow root vegetation to prepare for drought disasters in advance [59].

With a lagging effect, drought in winter and spring can directly affect the canopy growth and carbon balance in the early summer area, making the NDVI continuum in summer woodlands lower than the baseline (Figure 5e). Forest exhibits a higher ability to maintain fundamental metabolism and productivity in the face of a severe water shortage by changing stomatal openings to prevent excessive water loss [60,61]. The canopy of a forest is developed, giving it an advantage over cultivated land and grassland in terms of water vapor transmission and interception. The canopy increases evaporation at the canopy surface while successfully reducing evaporation of soil water. When precipitation drops, the moisture content of the air drops, the soil loses water, and tree roots reach deeper into the soil for water. However, as a drought deepens and persists, the deep soil moisture will decrease and the forest will not be able to obtain the water needed for normal growth. The growth status will then change significantly. For example, the leaf area and canopy coverage may decrease, and the photosynthetic capacity of the vegetation will be affected [62].

The mixed forest in the study area is mostly distributed at the forest edge, where the correlation between NDVI and SPEI is lower, but the response time scale is longer. This is due to the high canopy, the low variation range of temperature and surface temperature in the forest, low wind speed, low evaporation and mutual promotion of tree species that increase water use efficiency and improve the drought resistance of mixed forest. The MRYRB also contains a considerable number of broad-leaved forests, whose growth is mostly influenced by solar radiation, and the utilization rate of light energy increases the resistance of broad-leaved forests to drought [63]. When drought occurs, broadleaf forests remove the old leaves and keep the new ones, resulting in a high light use efficiency.

### 4.3. Impact of Ecological Project Construction on Vegetation Resistance

Substantial economic growth has resulted in serious ecological environmental contamination in the MRYRB. The rapid economic development and use of land have exacerbated the depletion of natural vegetation during the modern period. Furthermore, the region’s loess soil is readily eroded, with a low vegetation cover, and hence the ecological security of the study area cannot be ignored [64,65]. Since 1999, China has been carrying out the “Grain to Green” project in order to improve the natural environment, and this has caused the MRYRB to rapidly green-up (Figure 8).

Vegetation in 14.8% of the region has obviously improved (0.06 < *Slope*; *R* = 0.297), 25.8% of the areas have improved moderately (0.04 < *Slope* < 0.06; *R* = 0.305), 39.1% of the areas have improved slightly (0.02 < *Slope* < 0.04; *R* = 0.340), 20.1% of the areas have essentially remained unchanged (0 < *Slope* < 0.02; *R* = 0.358), only 0.2% of the areas have become degraded (*Slope* < 0; *R* = 0.146). These last areas are mainly distributed in areas with rapid urban development, and areas containing natural pastures such as Inner Mongolia. Animal husbandry is mainly distributed in central and western Inner Mongolia and northern Shaanxi. Here, overgrazing can lead to the degradation of grasslands, which can deteriorate when the number of livestock exceeds the carrying capacity of the grasslands. In this study, it was found that the maximum correlation coefficients were low in the areas with obvious vegetation improvement. An increase in forest area is one of the main reasons for the increase in NDVI as the high regional vegetation coverage results in low surface temperature and small evapotranspiration, so the response to drought is slow, and hence the maximum correlation coefficients of ecological improvement areas are lower.

The implementation of ecological engineering can promote the improvement of vegetation status, while the inappropriate occupation of cultivated land and the unplanned expansion of construction areas can lead to sporadic vegetation degradation in the study area [51,66]. These findings suggest that improving land-use efficiency, optimizing land use structures, strengthening cultivated land structure protection, promoting cultivated land structure improvement, and high-quality cultivated land construction, as well as alleviating the pressures of economic growth on the vegetation ecosystem, should be prioritized in future ecological restoration projects.

## 5. Conclusions

In this study, NDVI and SPEI time series were used to reveal the spatial distribution of the response of vegetation to drought in the MRYRB, analyze the temporal variation of resistance to drought, and verify the resistance duration of vegetation to drought in typical drought events. The main conclusions were as follows:
Drought intensity in the study area showed a weak increasing trend. The linear growth rates of SPEI-1, SPEI-3, SPEI-6, SPEI-9, and SPEI-12 were 0.002, 0.0034, 0.0198, 0.0234 and 0.0249, respectively.NDVI and SPEI were positively correlated in most areas (97.6%) of the MRYRB. Only 2.4% of the area showed negative correlations. Vegetation was most affected by drought on long-term time scales (9 and 12 months). The most influential time scale was shorter in areas with a higher maximum correlation coefficient.The resistance of vegetation decreased during drought, and gradually returned to its original level after the end of the drought. During a drought event, grassland and cultivated land (as shallow-rooted vegetations) had shorter resistance durations (i.e., around 170 days), while forest had a resistance duration of about 260 days.

Vegetation resistance to drought is caused by a variety of factors. This study focused on the resistance of different vegetation types, and the results suggest the important role of forests in delaying drought under the scenario of future global warming and frequent droughts.

## Figures and Tables

**Figure 1 ijerph-19-04180-f001:**
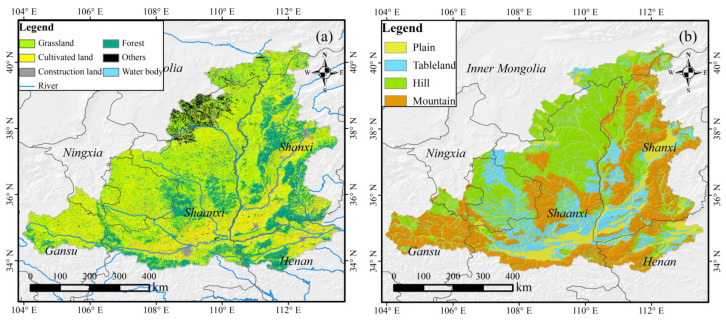
The spatial distribution of vegetation types (**a**) and geomorphic types (**b**) in the middle reaches of the Yellow River Basin (MRYRB).

**Figure 2 ijerph-19-04180-f002:**
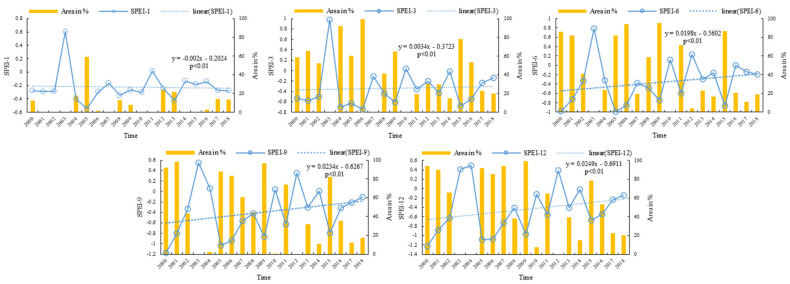
Temporal variation of SPEI at different time scales (blue line) and percentage of arid region areal cover (yellow bars) from 2000 to 2018. Time scales were 1 month (SPEI−1), 3 months (SPEI−3), 6 months (SPEI−6), 9 months (SPEI−9) and 12 months (SPEI−12).

**Figure 3 ijerph-19-04180-f003:**
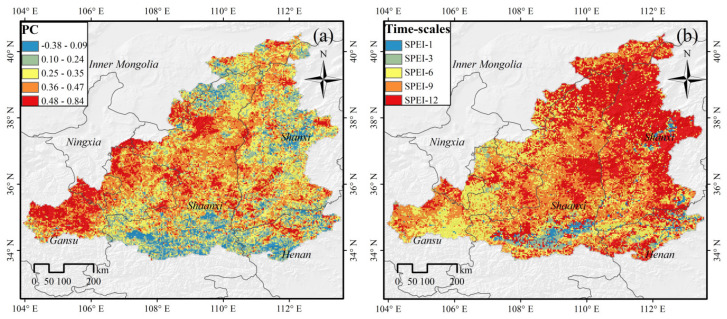
(**a**) The spatial pattern of the maximum correlation coefficient between NDVI and SPEI for the period of 2000–2018. (**b**) The time-scales at which the maximum correlation coefficient between NDVI and SPEI were obtained.

**Figure 4 ijerph-19-04180-f004:**
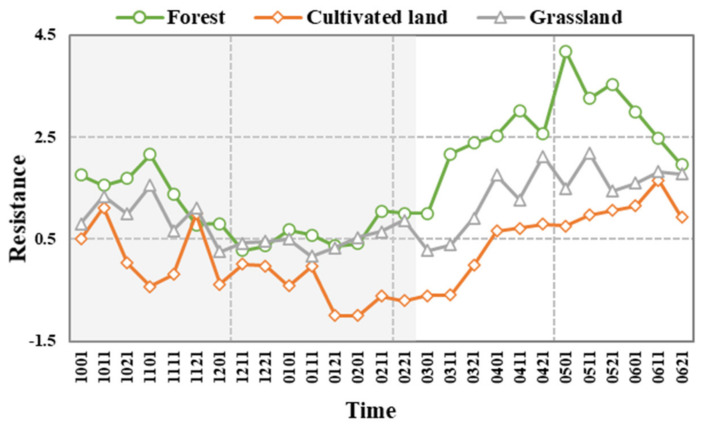
Temporal variation of drought resistance in three different vegetation types (forest, cultivated land, grassland) from October 2010 to June 2011. Time is expressed as month, day (i.e., 1001−1 October 2010; 0621−21 June 2011). Shading area indicates the period of drought. Droughts started in early October (1001), reached a peak in early February (0201) and began to ease in late February (0221).

**Figure 5 ijerph-19-04180-f005:**
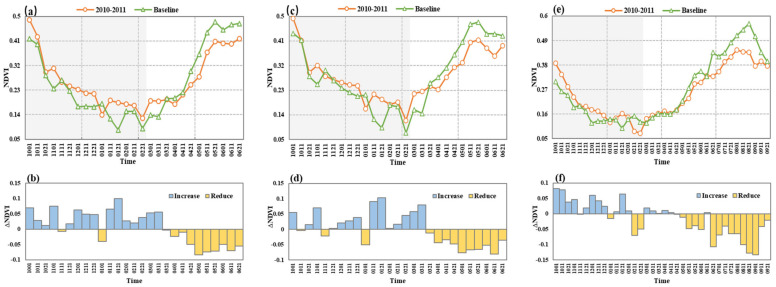
Temporal variation of NDVI in different vegetation types. (**a**,**b**) grassland, (**c**,**d**) cultivated land, (**e**,**f**) forest. (**a**,**c**,**e**) Variation of NDVI in 2010–2011 compared to baseline and (**b**,**d**,**f**) anomalies of NDVI in 2010–2011 relative to baseline. Time is expressed as month day (i.e., 1001−1 October 2010; 0621–21 June 2011). Shading area indicates the period of drought. Baseline condition is represented by the NDVI of 2011−2012.

**Figure 6 ijerph-19-04180-f006:**
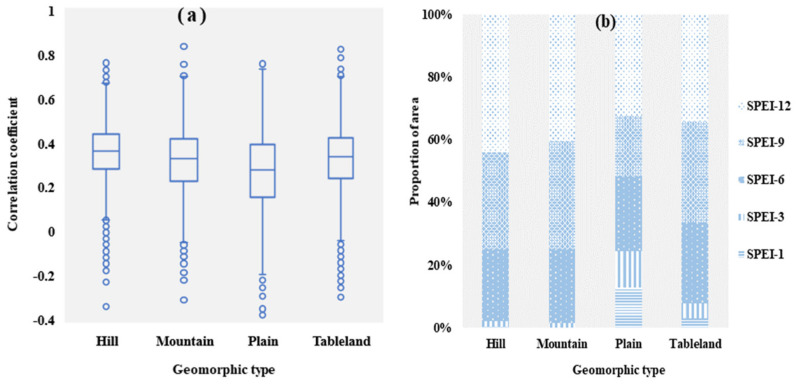
The maximum correlation coefficient of different geomorphic types: (**a**) Box−and−whisker plots (25th to 75th percentiles at the ends of the box. Median is indicated with a horizontal line in the interior of the box. Maximum and minimum values are at the ends of the whiskers); (**b**) Proportions with the maximum correlation at different time scales by geomorphic types.

**Figure 7 ijerph-19-04180-f007:**
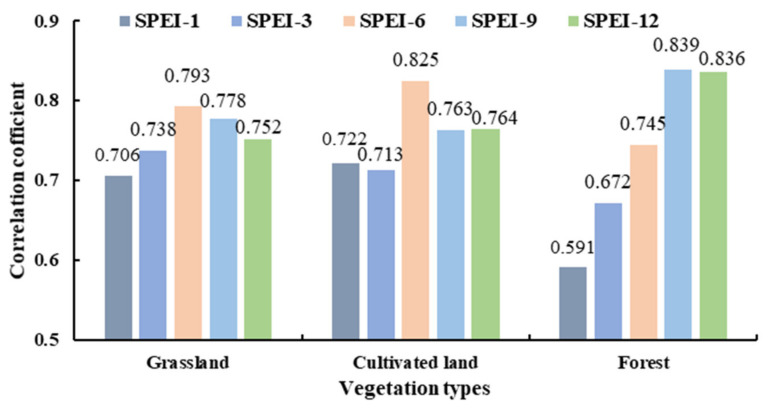
Maximum correlation coefficient between NDVI and SPEI for different vegetation types.

**Figure 8 ijerph-19-04180-f008:**
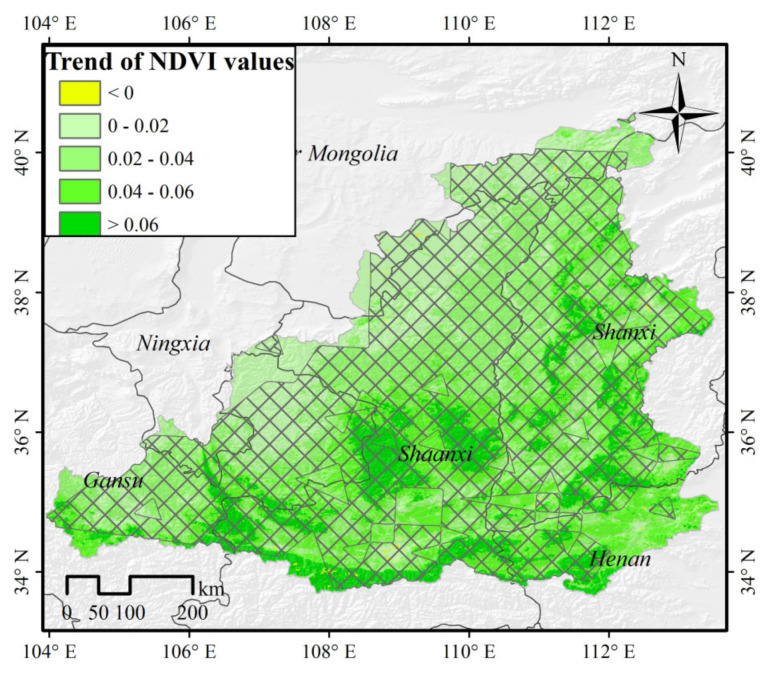
NDVI variation trend in the MRYRB from 2000 to 2018. The crosshatch indicates that the trend is statistically significant at the 95% confidence level based on *t*−test.

**Table 1 ijerph-19-04180-t001:** The information of datasets used in this study.

Name	Source	Time	Resolution
MODIS-NDVI	the US National Aeronautical Space Agency	2000–2018	1 km
SPOT-NDVI	Copernicus Global Land Service	2010–2012	1 km
Land-cover	Resource and Environment Science and Data Cloud Platform	2018	1 km
Geomorphic type	Chinese 1:1,000,000 Geomorphic Atlas	-	-
DEM	Geospatial Data Cloud	-	90 m
SPEI	Climate Research Unit	2000–2018	0.5° × 0.5°

## Data Availability

Not applicable.

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
