# Peer review of "Assessing Vegetation Ecosystem Resistance to Drought in the Middle Reaches of the Yellow River Basin, China"

_ijerph, 2022, doi:10.3390/ijerph19074180_

Round 1

Reviewer 1 Report

The authors of this article deal with very interesting topics, clearly explaining the used data and methods, describing the results and discussions. In addition, the introduction and method sections could be improved by specifying some important topics such as the global warming trend affecting several ecosystems, the SPEI, and further vegetation data used in the study.  

Following some minor comments helpful to the authors.

  1. It is advisable to specify (in the entire manuscript) what kind of vegetation was analyzed in the study (grass, trees, bushes) and the data used for reference to it.
  2. In the introduction, it is advisable to describe some important topics such as the increasing global warming trend affecting ecosystems, the importance of monitoring land surface temperature and vegetation elements,  by mentioning some recent studies that have applied different methods such as the Getis-Ord Gi* statistics. The following studies are suggested: https://doi.org/10.1038/nclimate2067, https://doi.org/10.1038/s41612-021-00218-2, https://doi.org/10.1016/j.ecoinf.2019.101032.
  3. In the introduction, It would be great to briefly describe SPEI index and indicate some recent studies which used this index by applying a methodological approach similar to your research. Some suggested studies on this topic are the following: https://doi.org/10.1007/s13762-021-03852-8,https://doi.org/10.1016/j.envc.2021.100410.
  4. In the methods section please add some references regarding the vegetation resistance index formula and its application to recent studies.

Author Response

Response to Reviewer 1 Comments
Point 1: It is advisable to specify (in the entire manuscript) what kind of vegetation was analyzed in the study (grass, trees, bushes) and the data used for reference to it.
Response 1: Thanks very much for your valuable comments and suggestion; they are very helpful for us. Your affirmation is a great encouragement to us. In the manuscript, we have added the vegetation types analyzed in the study. In section 3.3 and section 4.2, we revealed the differences in resistance and resistance duration and reasons for the three main vegetation types: forest, cultivated land, and grassland.
Modifications: L150:” The obtained land-cover data are summarized and reclassified into 6 categories: grassland, forest, cultivated land, construction land, water body and others (Fig. 1a). The study in this paper focuses on grassland, forest, and cultivated land.”added.

Point 2: In the introduction, it is advisable to describe some important topics such as the increasing global warming trend affecting ecosystems, the importance of monitoring land surface temperature and vegetation elements, by mentioning some recent studies that have applied different methods such as the Getis-Ord Gi* statistics. The following studies are suggested:
https://doi.org/10.1038/nclimate2067,https://doi.org/10.1038/s41612-021-00218-2, https://doi.org/10.1016/j.ecoinf.2019.101032.

Response 2: Thank you for your scrupulous reminding. We accepted it and revised it.
Modifications:
L30: ”Human activities cause an increase in the concentration of carbon dioxide and other heattrapping gases in the atmosphere, which in turn leads to an increase in surface temperature and global warming, and the frequency and intensity of droughts will continue to increase by the end of the 21st century, especially in arid regions [2,3]. The composite drought hot spot map by Samiul et al. [4] draws attention to the negative effects of drought”added.
L40: ”Vegetation provides important ecosystem services related to human well-being, biodiversity and carbon cycling, and is an important component of terrestrial ecosystems [6,7]. Vegetation is characterized by adaptation to its environment, and therefore, close monitoring of vegetation spectral characteristics is of ecological importance.”added L531: “2. Trenberth, K.E.; Dai, A.; Schrier, G.V.D.; Jones, P.D.; Barichivich, J.; Briffa, K.R.; Sheffield, J. Global warming and changes in drought. Nat. Clim. Change 2014, 4, 17-22.3. Balting., D.F.; Amir, A.; Gerrit, L.; Monica, I. Northern Hemisphere drought risk in a warming climate. npj Clim. Atmos.Sci. 2021, 4, 1-13. 4. Samiul, I.S.M.; Ashraful, I.K.M.; Akter, M.M.R. Drought hot spot analysis using local indicators of spatial autocorrelation: An experience from Bangladesh. Environ. Challenges 2022, 6, 100410.”added. 2 L538: “6. Cârlan, I.; Mihai, B.-A.; Nistor, C.; Große-Stoltenberg, A. Identifying urban vegetation stress factors based on open access remote sensing imagery and field observations. Ecol. Inform. 2020, 55, 101032. 7. Sugiyama, T.; Carver, A.; Koohsari, M.J.; Veitch, J. Advantages of public green spaces in enhancing population health. Landscape. Urban Plan. 2018, 178, 12-17.”added.

Point 3: In the introduction, It would be great to briefly describe SPEI index and indicate some recent studies which used this index by applying a methodological approach similar to your research. Some
suggested studies on this topic are the following: https://doi.org/10.1007/s13762-021-03852- 8,https://doi.org/10.1016/j.envc.2021.100410.
Response 3: Thank you for your scrupulous reminding. We accepted it and revised it.
Modifications:
L56:” SPEI considers both temperature and precipitation and is able to detect the effects of evapotranspiration and temperature changes on global warming [17].”added.
L59:” For example, SPEI was analyzed by Nejadrekabi et al. [20] using the Getis-Ord-Gi statistics to determine the SPEI values that form clusters and identify areas at high risk of drought; Sharma et al. [17] used geospatial techniques to generate the SPEI drought map of Tripura, Northeast India in 1985 to describe the severity of the drought.”added.
L562:”17. Sharma, A.P.M.; Jhajharia, D.; Yurembam, G.S.; Gupta, S. Assessment of
Meteorological Drought with Application of Standardized Precipitation Evapotranspiration Index (SPEI) for Tripura, Northeast India. Int. J Environ. Clim.Change 2021, 11, 126-135.” added.
L570:”20. Nejadrekabi, M.; Eslamian, S.; Zareian, M.J. Spatial statistics techniques for SPEI and NDVI drought indices: a case study of Khuzestan Province. Int. J. Environ. Sci. Te 2022, 1-22.”added.

Point 4: In the methods section please add some references regarding the vegetation resistance index formula and its application to recent studies.
Response 4: Thank you for your scrupulous reminding. We accepted it and revised it. Regarding the calculation of the vegetation resistance index, the previous method is followed in this paper, and the specific calculation steps can be seen in Equation (1). Li et al. used the above formula in study of the
sensitivity of gymnosperms to extreme drought and Isbell et al. in revealing the effect of biodiversity on ecosystem resistance, both of which have been cited in the method section.
Modifications:
L610: “39. Li, X.; Piao, S.; Wang, K.; Wang, X.; Wang, T.; Ciais, P.; Chen, A.; Lian, X.; Peng, S.; Peñuelas, J. Temporal trade-off between gymnosperm resistance and resilience increases forest sensitivity to extreme drought. Nat. Ecol. Evol. 2020, 4, 1075-1083. 40. Isbell, F.; Craven, D.; Connolly, J.; Loreau, M.; Schmid, B.; Beierkuhnlein, C.; Bezemer, T.M.; Bonin, C.; Bruelheide, H.; Enrica, D.L.; et al. Biodiversity increases the resistance of ecosystem productivity to climate extremes. Nature 2015, 526, 574-577.”added.

Thanks again for your good ideas and suggestion, we appreciate it very much.

Reviewer 2 Report

The subject is interesting to readers, but in my opinion the manuscript has many issues that need to be solved.

My observations are briefly described below.

L29-30: Is the drought a hydrological event? Does the lack of water only cause drought? The definition of drought is much more complex and depends on the type of drought referred to. See Wilhite, D.A .; and M.H. Glantz. 1985. Understanding the Drought Phenomenon: The Role of Definitions. Water International 10 (3): 111–120. Please reformulate!

L49: SPEI is used in climatology by climatologists, not in meteorology by meteorologists. Please correct!

L124-126: Please specify the name of the product used!

L128: What does "De-clouded" mean? Do you mean cloud free? Please reformulate!

L124-133: Why did you choose to use NDVI from both MODIS and SPOT? Both products are 1 km away, both cover the same time frame, I don't see the point of using 2 data sets for the same product. Please explain!

I also recommend making a table to summarize the data sets used. Why did the analysis stop in 2018? What is the complete analysis period, 1989-2018 or 2000-2018? I would recommend that the analysis be done by 2020 at least!

https: // lad-126 sweb.modaps.eosdis.nasa.gov and http://www.VEG.vito.be not opening! Please specify the date of access! Did you only use data from Spot-Vegetation? Why didn't you continue with the Proba Vegetation (which is a continuation of the Spot-Vegetation mission)?

L:145-150: What is the time step used for SPEI, 1 month, 3 months, 6 months 12 months? Please specify! Also, argue the choice made!

L153-169: Is this method of analyzing the resistance of vegetation proposed by the authors and is it a new method or has it been applied in other studies? If it has been applied before, please quote the respective works or specify that it is a new method, proposed by the authors.

L171-193: Pearson correlation is not a new method, but I do not see any cited source! Do the authors assume these methods to be proposed by them?

L196-200: In the methodology you presented the use of the slope for the analysis of the NDVI trend, and you did not say anything about its use in the trend analysis of the SPEI index, please complete! You also gave the slope values, but what do these values mean, are they statistically significant or not? Have you applied any statistical tests to see if the date is significant?

L206-208: Do these claims about climate change have a scientific basis? Please specify the source!

L214-223: The correlation is made between NDVI and SPEI for what time step? NDVI from which data set, MODIS or Spot?

L224-226: What type of vegetation has been affected by the annual drought? What does the annual period mean, from January 1st to December 31st? Is this interval suitable for the analysis of all types of vegetation? In this case, what was the annual drought related to, NDVI at the annual or monthly level? Specify the time step for NDVI!

L224-233: What are the months for the time steps of 1, 3, 6, 9, 12 months mentioned in the case of SPEI? What are the time steps for NDVI and what are the months for NDVI?

L239-245: Is the information in this paragraph obtained by the authors based on their own data processing or from various sources?

L322-323: “According to previous studies ", but you have only one quoted study, are there several or only one that makes this statement?

L323-328: Were these aspects studied in this manuscript or are the information taken from other sources?

Author Response

Response to Reviewer 2 Comments

Point 1: L29-30: Is the drought a hydrological event? Does the lack of water only cause drought? The definition of drought is much more complex and depends on the type of drought referred to. See Wilhite, D.A .; and M.H. Glantz. 1985. Understanding the Drought Phenomenon: The Role of Definitions. Water International 10 (3): 111–120. Please reformulate!

Response 1: Thanks very much for your valuable comments and suggestion; they are very helpful for us. Your affirmation is a great encouragement to us. We accepted it and revised it.

Modifications:

L29: ” Drought is a condition relative to some long-term average condition of balance between rainfall and evapotranspiration in a particular area [1].”added.

L529: ”1. Wilhite, D.A.; Glantz, M.H. Understanding: the Drought Phenomenon: The Role of Definitions. Water Int. 1985, 10, 111-120.”added.

Point 2: L49:SPEI is used in climatology by climatologists, not in meteorology by meteorologists. Please correct!

Response 2: Thank you for your scrupulous reminding. We accepted it and revised it by replacing " widely used by meteorologists in meteorology research " to "widely used by climatologists in climatology research " in L58.

Point 3: L124-126:Please specify the name of the product used!

Response 3: Thank you for your scrupulous reminding. We accepted it.

Modifications:

L137: “(MOD13A3)”added.

L140:” (SPOT-VEGETATION NDVI)”added.

Point 4: L128: What does "De-clouded" mean? Do you mean cloud free? Please reformulate!

Response 4: Thank you for your scrupulous reminding. "De-clouded" refers to the SPOT-NDVI dataset that has undergone removal of cloud effects. The remote sensing images are covered by clouds during the imaging process. Before distributing to users, the data production unit performs radiation correction, geometric correction and atmospheric correction on SPOT-NDVI data to remove the influence of clouds, ice and snow to ensure the quality of the data [1,2,3].

[1]Girma A, de Bie C, Skidmore A K, et al. Hyper-temporal SPOT-NDVI dataset parameterization captures species distributions[J]. International Journal of geographical information Science, 2016, 30(1): 89-107.

[2]He Z, He J. Spatio-temporal variation of vegetation cover based on SPOT-VGT in Yellow River Basin [J]. Ecology and Environmental Sciences, 2012, 21(10): 1655-1659. (in Chinese) 2

[3]An Z, Gao W, Gao Z, et al. Trend analysis for evaluating the consistency of Terra MODIS and SPOT VGT NDVI time series products in China[J]. Frontiers of Earth Science,2015,9(1):125-136.

Modifications:

L139:” The 10-day maximum SPOT-NDVI (SPOT-VEGETATION NDVI) synthesis images at 1 km spatial resolution were obtained from the Resource and Environment Science and Data Cloud Platform (http://www.resdc.cn/) for 2010-2012, which has removed the effects of clouds, ice and snow [35].”revised.

L602:”35. An, Y.; Gao, W.; Gao, Z.; Liu, C.; Shi, R. Trend analysis for evaluating the consistency of Terra MODIS and SPOT VGT NDVI time series products in China. Front. Earth Sci. 2015, 9, 125-136.”added.

Point 5: L124-133:Why did you choose to use NDVI from both MODIS and SPOT? Both products are 1 km away, both cover the same time frame, I don't see the point of using 2 data sets for the same product. Please explain!

Response 5: Thank you for your scrupulous reminding. The temporal resolution of MODIS is 1 month, which is used in the study to calculate the correlation coefficient with the drought index SPEI, and the correlation coefficient calculation does not require much temporal resolution of the product data; while the temporal resolution of SPOT is 10 days (1st-10th, 11th-20th, 21st - end of each month), which is used in the paper to calculate the vegetation resistance and analyze the vegetation resistance duration for the drought event (2010.10-2011.02), and the higher temporal resolution helps to present the differences between different vegetation types in more detail. The use of the two NDVI product data is described in L143.

Point 6: I also recommend making a table to summarize the data sets used. Why did the analysis stop in 2018? What is the complete analysis period, 1989-2018 or 2000-2018? I would recommend that the analysis be done by 2020 at least!

Response 6: Thank you for your scrupulous reminding. We accepted it. SPEI product data are obtained from the Climate Research Unit for the period 2000-2018, and the latest data have not been updated, therefore, the complete analysis period from 2000-2018.

Modifications:

L168: “Table 1.”added.

Point 7: https: // lad-126 sweb.modaps.eosdis.nasa.gov and http://www.VEG.vito.be not opening! Please specify the date of access! Did you only use data from Spot-Vegetation? Why didn't you continue with the Proba Vegetation (which is a continuation of the Spot-Vegetation mission)?

Response 7: Thank you for your scrupulous reminding. The MODIS dataset was obtained from https://ladsweb.modaps.eosdis.nasa.gov, accessed in March 2020, can be opened normally at present. The SPOT dataset was obtained from http://www.VEG.vito.be, accessed in April 2020, but the website is now closed, and the same dataset is available from the Resource and Environment Science 3

and Data Cloud Platform (http://www.resdc.cn/). Therefore, we have changed the website where the data is available in the manuscript so that readers can download the dataset in L142.

SPOT-NDVI was used in the manuscript for vegetation resistance analysis for typical drought events. The drought events selected in the China Drought Disaster Bulletin were 2010.10-2011.02, and the available SPOT-NDVI data were sufficient to meet the demand. In the manuscript, we modified the time span of the acquired SPOT-NDVI in L142. In other subsequent studies, I will consider using the Proba Vegetation dataset as a supplement to Spot-Vegetation.

Point 8: L:145-150: What is the time step used for SPEI, 1 month, 3 months, 6 months 12 months? Please specify! Also, argue the choice made!

Response 8: Thank you for your scrupulous reminding. We accepted it and revised it The product data of the downloaded SPEI include 1, 3, 6, 9, and 12 months time scales. The n-month SPEI includes

the cumulative climate water balance of the previous n months (including the current month), which can be used to identify droughts at different time scales [1]. For example, the 1- month SPEI represents the drought conditions in the current month, and the 12-month SPEI provides the drought conditions for the whole year.

Most of the current studies on the response of vegetation to drought suggest that the response time is concentrated in the period of 1-12 months [2-4]. Therefore, in the manuscript, we use the SPEI at time scale of 1 and 3 months to characterize the short-term dry and wet conditions, the SPEI at time scale of 6 months to characterize the medium-term dry and wet conditions, and the SPEI at time scale of 9 and 12 months to characterize the long-term dry and wet conditions.

[1]Wei X,He W,Zhou Y,et al. Global assessment of lagged and cumulative effects of drought on grassland gross primary production[J]. Ecological Indicators,2022,136:108646.

[2]Kang W, Wang T, Liu S. The response of vegetation phenology and productivity to drought in semi-arid regions of Northern China[J]. Remote Sensing, 2018, 10(5): 727.

[3]Yang S, Meng D, Li X, et al. Multi-scale responses of vegetation changes relative to the SPEI meteorological drought index in North China in 2001-2014[J]. Acta Ecologica Sinica,2018,38(3): 1028-1039.(in Chinese).

[4]Khatri-Chhetri P, Hendryx S M, Hartfield K A, et al. Assessing vegetation response to multi-scalar drought across the mojave, sonoran, chihuahuan deserts and apache highlands in the Southwest United States[J]. Remote Sensing, 2021, 13(6): 1103.

Modifications:

L160:” SPEI product datasets for the time scales of 1, 3, 6, 9 and 12 months came from the Climate Research Unit (CRU) (https://digital.csic.es/handle/10261/202305) with a spatial resolution of 0.5° x 0.5° for the period 2000-2018, representing short (1 and 3 months), medium (6 months), and long (9 and 12 months) time scales, respectively.”added.

Point 9: L153-169: Is this method of analyzing the resistance of vegetation proposed by the authors and is it a new method or has it been applied in other studies? If it has been applied before, please quote the respective works or specify that it is a new method, proposed by the authors. 4

Response 9: Thank you for your scrupulous reminding. We accepted it and revised it. The calculation method of vegetation resistance index has been widely used in previous studies, and relevant references have been added. Vegetation resistance duration is a new method, and the specific definition is described in detail in L183.

Modifications:

L610: “39. Li, X.; Piao, S.; Wang, K.; Wang, X.; Wang, T.; Ciais, P.; Chen, A.; Lian, X.; Peng, S.; Peñuelas, J. Temporal trade-off between gymnosperm resistance and resilience increases forest sensitivity to extreme drought. Nat. Ecol. Evol. 2020, 4, 1075-1083.

  1. Isbell, F.; Craven, D.; Connolly, J.; Loreau, M.; Schmid, B.; Beierkuhnlein, C.; Bezemer, T.M.; Bonin, C.; Bruelheide, H.; Enrica, D.L.; et al. Biodiversity increases the resistance of ecosystem productivity to climate extremes. Nature 2015, 526, 574-577.”added.

Point 10: L171-193: Pearson correlation is not a new method, but I do not see any cited source! Do the authors assume these methods to be proposed by them?

Response 10: Thank you for your scrupulous reminding. We accepted it and revised it. Pearson correlation analysis, Maximum Value Composite method, and trend analysis are not new methods and have been widely used in related studies and have been supplemented with the most recent references in the corresponding paragraphs of the manuscript.

Modifications:

L556. :”14. Yin, J.; Yuan, Z.; Li, T. The Spatial-Temporal Variation Characteristics of Natural Vegetation Drought in the Yangtze River Source Region, China. Int. J. Environ. Res. Public Health 2021, 18, 1613.”added.

L618:”42. Wang, Y.; Fu, B.; Liu, Y.; Li, Y.; Feng, X.; Wang, S. Response of vegetation to drought in the Tibetan Plateau: Elevation differentiation and the dominant factors. Agr. Forest Meteorol. 2021, 306, 108468.

  1. Zhu, X.; Liu, Y.; Xu, K.; Pan, Y. Effects of Drought on Vegetation Productivity of Farmland Ecosystems in the Drylands of Northern China. Remote Sens. 2021, 13, 1179.”added.

Point 11: L196-200: In the methodology you presented the use of the slope for the analysis of the NDVI trend, and you did not say anything about its use in the trend analysis of the SPEI index, please complete! You also gave the slope values, but what do these values mean, are they statistically significant or not? Have you applied any statistical tests to see if the date is significant?

Response 11: Thank you for your scrupulous reminding. We accepted it and revised it. Regarding the calculation of the SPEI index has been added in the methods section, slope values greater than 0 imply an upward trend of the variable and slopes less than 0 imply a downward trend of the variable, and a significance test was performed using Excel software. Significance tests regarding NDVI trends have been added in Figure 8, where the crosshatch indicates that the trend is statistically significant at the 95% confidence level based on T-test. And 83.1% of the regions passed the significance at the 95% confidence level.

Modifications:

L189:” To calculate the trend in variable using the linear tendency method [40]. In this study, the variable is SPEI. The formula is as follows: 5

(3) = +y abx

where y is the time series of SPEI, x is the corresponding time series; the regression coefficient b represents the linear trend, and a is a constant. The value of b greater than 0 implies an increasing trend.” added.

L224:” all of which passed the significance test (p<0.01).” added.

L616:”41. Ma, S.; Zhou, T.; Dai, A.; Han, Z. Observed Changes in the Distributions of Daily Precipitation Frequency and Amount over China from 1960 to 2013. J. Climate 2015, 28, 6960- 6978.”added.

Point 12: L206-208: Do these claims about climate change have a scientific basis? Please specify the source!

Response 12: Thank you for your scrupulous reminding. Liu et al [1] used 507 station observations from the CMIP5 multi-model integration to predict China's climate change trends under RCP2.6, RCP4.5, and RCP8.5 scenarios, respectively. The climate in China changed from cold to warm in the last half-century, accompanying the transformation of Cold-Humid, Cold-Dry, and Cold-Normal before the early 1990s to Warm-Humid, Warm-Dry, and Warm-Normal from the early 1990s onward. In the 21st century, the projected climatic year types are mainly Warm-Humid, Warm-Dry, and Warm-Normal in China. The conclusion is consistent with the alternating wet and dry climate in the MRYRB in this section, which has been cited in L236.

[1]Liu, J.; Du, H.; Wu, Z.; He, H.; Wang, L.; Zong, S. Recent and future changes in the combination of annual temperature and precipitation throughout China. Int. J. Climatol. 2017, 37, 821-833.

Modifications:

L232: “This is consistent with the observation of Liu et al. that warm-dry is the dominant climate feature in the region, and that alternating high-frequency warm-humid, warm-dry, and warm-normal changes will become the mainstream climate feature in the 21st century under the projected conditions of climate change scenarios RCP2.6, RCP4.5, and RCP8.5 [33]. ” revised.

Point 13: L214-223: The correlation is made between NDVI and SPEI for what time step? NDVI from which data set, MODIS or Spot?

Response 13: Thank you for your scrupulous reminding. Based on the Maximum Value Composite method, the monthly MODIS-NDVI were synthesized on an annual scale and correlated with different time scales SPEI for 19 years from 2000-2018, where NDVI data were obtained from the MODIS dataset. We revised it by replacing " NDVI " to "MODIS-NDVI " in L242.

Point 14: L224-226: What type of vegetation has been affected by the annual drought? What does the annual period mean, from January 1st to December 31st? Is this interval suitable for the analysis of all types of vegetation? In this case, what was the annual drought related to, NDVI at the annual or monthly level? Specify the time step for NDVI!

Response 14: Thank you for your scrupulous reminding. In section 3.2, we analyzed the correlation between annual scale NDVI and SPEI at the regional scale, and did not specifically distinguish different vegetation types. In section 4.2, we discussed the differences in the response of different vegetation types to drought, and found that forest were mainly influenced by SPEI at the time scales 6

of 9 and 12 months. the SPEI at the time scales of 12 months provides the drought conditions for the whole year. The correlation coefficient is calculated with SPEI using the annual NDVI.

Modifications:

L254.:We revised it by replacing " indicating that the MRYRB was most affected by annual drought " to “indicating that the annual NDVI of MRYRB was most affected by SPEI at time scale of 12 months (SPEI-12)” with red highlight.

Point 15: L224-233: What are the months for the time steps of 1, 3, 6, 9, 12 months mentioned in the case of SPEI? What are the time steps for NDVI and what are the months for NDVI?

Response 15: Thank you for your scrupulous reminding.The time step of SPEI is 1 year for different time scales, SPEI and MODIS-NDVI were used to synthesize annual data using the mean method and the Maximum Value Composite method, respectively, to represent the dry and wet conditions and vegetation cover levels in that year, and to calculate the annual correlation coefficients.

Point 16: L239-245:Is the information in this paragraph obtained by the authors based on their own data processing or from various sources?

Response 16: Thank you for your scrupulous reminding. This information is obtained from the Bulletin of Flood and Drought Disasters in China of the Ministry of Water Resources of the People's Republic of China (http://www.mwr.gov.cn/), which is cited in L268.

Modifications:

L624:”45. State Flood Controland Drought Relief Headquarters, Ministry of Water Resources of the People’s Republic of China. Bulletin of Flood and Drought Disaster in China (2011); China Waterpower Press: Beijing, China, 2012.”added.

Point 17: L322-323: “According to previous studies ", but you have only one quoted study, are there several or only one that makes this statement?

Response 17: Thank you for your scrupulous reminding. We accepted it. There exists several literatures with this view, which is added in the manuscript.

Modifications:

L568:” 19. Zhao, A.; Zhang, A.; Cao, S.; Liu, X.; Liu, J.; Cheng, D. Responses of vegetation productivity to multi-scale drought in Loess Plateau, China. Catena 2018, 163, 165-171.”added.

L629:” 47. Herrmann, S.M.; Didan, K.; Barreto-Munoz, A.; Crimmins, M.A. Divergent responses of vegetation cover in Southwestern US ecosystems to dry and wet years at different elevations. Environ. Res. Lett. 2016, 11, 124005.”added.

Point 18: L323-328: Were these aspects studied in this manuscript or are the information taken from other sources? 7

Response 18: Thank you for your scrupulous reminding. This information was summarized by reviewing materials, reading literature and books, and has been supplemented with corresponding references.

Modifications:

L634:”49. Yang, Q.; Li, Z.; Han, Y.; Gao, H. Responses of Baseflow to Ecological Construction and Climate Change in Different Geomorphological Types in The Middle Yellow River, China. Water 2020, 12, 304.

  1. Bao, Z.; Zhang, J.; Wang, G.; Chen, Q.; Guan, T.; Yan, X.; Liu, C.; Liu, J.; Wang, J. The impact of climate variability and land use/cover change on the water balance in the Middle Yellow River Basin, China. J. Hydrol. 2019, 577, 123942.”added.

Thanks again for all of your good ideas and suggestion, we appreciate it very much.

Round 2

Reviewer 1 Report

Thanks to the authors for having made all the changes to the manuscript in detail.

Author Response

Thanks again for your good ideas and suggestions, they are very helpful for us. Your affirmation is a great encouragement to us. 

Reviewer 2 Report

Thanks to the authors for their answers. Most of the changes I have requested have been made, but I still have a recommendation that I believe should be made before the manuscript can be published.

The authors provided the source of the product SPOT-VEGETATION NDVI, the Resource and Environment Science and Data Cloud Platform (http://www.resdc.cn/), and their argument regarding the use of this data source is as follows: “The SPOT dataset was obtained from http://www.VEG.vito.be, accessed in April 2020, but the website is now closed, and the same dataset is available from the Resource and Environment Science and Data Cloud Platform (http://www.resdc.cn/). Therefore, we have changed the website where the data is available in the manuscript so that readers can download the dataset in L142.”

If the product SPOT VEGETATION NDVI on Resource and Environment Science and Data Cloud Platform (http://www.resdc.cn/) is identical to the one previously used on http://www.VEG.vito.be, I strongly recommend that the source of this product be Copernicus Global Land Service (https://land.copernicus.eu/global/products/ndvi or https://land.copernicus.vgt.vito.be/PDF/portal/Application.html#Home). It is the current official platform where this product can be downloaded.

Author Response

Response to Reviewer 2 Comments

Point 1: The authors provided the source of the product SPOT-VEGETATION NDVI, the Resource and Environment Science and Data Cloud Platform (http://www.resdc.cn/), and their argument regarding the use of this data source is as follows: “The SPOT dataset was obtained from http://www.VEG.vito.be, accessed in April 2020, but the website is now closed, and the same dataset is available from the Resource and Environment Science and Data Cloud Platform (http://www.resdc.cn/). Therefore, we have changed the website where the data is available in the manuscript so that readers can download the dataset in L142.”

If the product SPOT VEGETATION NDVI on Resource and Environment Science and Data Cloud Platform (http://www.resdc.cn/) is identical to the one previously used on http://www.VEG.vito.be, I strongly recommend that the source of this product be Copernicus Global Land Service (https://land.copernicus.eu/global/products/ndvi or https://land.copernicus.vgt.vito.be/PDF/portal/Application.html#Home). It is the current official platform where this product can be downloaded.

Response 1: Thanks very much for your valuable comments and suggestion, they are very helpful for us. Your affirmation is a great encouragement to us. We accepted it and revised it.

Modifications:

L139: ” The 10-day maximum SPOT-NDVI (SPOT-VEGETATION NDVI) synthesis images at 1 km spatial resolution were obtained from the Copernicus Global Land Service (https://land.copernicus.eu/global/products/ndvi) for 2010-2012, which has removed the effects of clouds, ice and snow.”reviseded.

L168: ” Table 1” reviseded.

Thanks again for your good ideas and suggestion, we appreciate it very much.